# Generalization v.s. Memorization: Tracing Language Models' Capabilities Back to Pretraining Data

**Xinyi Wang**[1*]**, Antonis Antoniades**[1*]**Yanai Elazar**[2,3]**, Alfonso Amayuelas**[1]**, Alon Albalak**[4]**,
Kexun Zhang**[5]**, William Yang Wang**[1]
[1]University of California, Santa Barbara, [2]Allen Institute for AI,
[3]University of Washington, [4] SynthLabs, [5]Carnegie Mellon University
{xinyi_wang,antonis}@ucsb.edu, william@cs.ucsb.edu

## Abstract

The impressive capabilities of large language models (LLMs) have sparked debate over whether these models genuinely generalize to unseen tasks or predominantly rely on memorizing vast amounts of pretraining data. To explore this issue, we introduce an extended concept of memorization, **distributional memorization**, which measures the correlation between the LLM output probabilities and the pretraining data frequency. To effectively capture task-specific pretraining data frequency, we propose a novel **task-gram language model**, which is built by counting the co-occurrence of semantically related $n$-gram pairs from task inputs and outputs in the pretraining corpus. Using the Pythia models trained on the Pile dataset, we evaluate four distinct tasks: machine translation, factual question answering, world knowledge understanding, and math reasoning. Our findings reveal varying levels of memorization, with the strongest effect observed in factual question answering. Furthermore, while model performance improves across all tasks as LLM size increases, only factual question answering shows an increase in memorization, whereas machine translation and reasoning tasks exhibit greater generalization, producing more novel outputs. This study demonstrates that memorization plays a larger role in simpler, knowledge-intensive tasks, while generalization is the key for harder, reasoning-based tasks, providing a scalable method for analyzing large pretraining corpora in greater depth.[1]

## 1 Introduction

Large language models (LLMs), such as GPT-4, have achieved remarkable performance across a wide range of tasks, yet the debate persists regarding whether these models are truly generalizing to unseen test cases or merely memorizing their extensive training data (Magar and Schwartz, 2022; Srivastava et al., 2024; Bender et al., 2021; Merrill et al., 2024; Shaib et al., 2024). Previous research has primarily investigated memorization in LLMs through verbatim recall of long segments from the training corpus (Zhang et al., 2023; Jiang et al., 2024; Carlini et al., 2022). However, the exact reproduction of long text is relatively uncommon when examining high-level capabilities such as translation and reasoning, particularly when the task output is short, as with world knowledge questions. To advance this line of inquiry, a more flexible definition of memorization is necessary.

Several works have explored the relationship between memorization and generalization (Feldman, 2020; Feldman and Zhang, 2020; Zhang et al., 2023), often employing *counterfactual memorization*, which measures the difference in model performance when a specific training example is excluded. However, these studies typically use small models and datasets, or subsets of larger datasets, and focus primarily on quantifying how much model behavior depends on memorization. They do not fully examine how different model capabilities emerge from the interplay between memorization and

---

[*]denotes equal contribution.
[1]Our code is available at: https://github.com/a-antoniades/llm-corpus-search

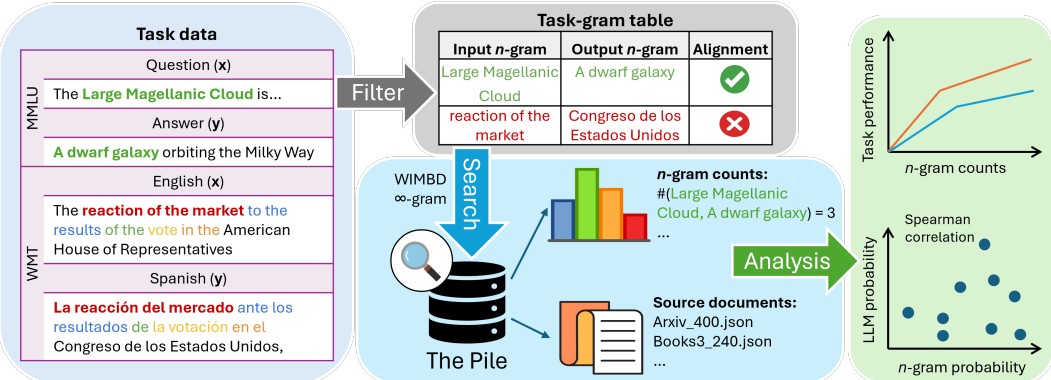

Figure 1: Overview of our proposed analysis pipeline. For the selected evaluation tasks, we first construct a **task-gram table** by matching semantically similar $n$-grams from task inputs ($x$) and targets ($y$). These $n$-grams are then searched within the pretraining corpus, yielding their counts and source documents. We then build a **task-gram language model** from the obtained counts and then analyze their relationship with LLM predictions.

generalization. Moreover, the limited scale of these analyses is partly constrained by the definition of counterfactual memorization, which requires expensive model retraining.

In this paper, we introduce a new framework for understanding memorization at scale, enabling us to analyze various LLM capabilities. We define **distributional memorization** as the correlation between the distribution of LLM outputs and the distribution of pretraining data. Similarly, we define **distributional generalization** as the divergence between the LLM's output distribution and the pretraining data distribution. For simplicity, we will refer to these concepts as *memorization* and *generalization* throughout the paper.

Our approach to defining memorization and generalization requires estimating the distribution of LLMs' pretraining corpora, which is a challenging task given the sheer size of these datasets, often containing trillions of tokens. To address this, we propose a novel method for modeling language distributions by counting semantically related $n$-gram pairs extracted from a task's input-output pairs. For example, in machine translation, these $n$-grams would correspond to phrase pairs from the source and target languages, as shown in Figure 1. This allows us to construct a set of $n$-gram pairs that characterize the task. Drawing inspiration from the *phrase table* used in machine translation (Passban et al., 2016), which consists of translation pairs, we refer to this set as the task's **task-gram table**. When $n$-grams from the input and output co-occur within a document, they are often separated by significant distances. By counting these co-occurrences, we are able to model long-range, task-relevant dependencies Elazar et al. (2019; 2022). The resulting $n$-gram language model, built from the task-gram table, is referred to as a **task-gram language model**. In contrast, classical $n$-gram LMs capture only local lexical dependencies within a single $n$-gram. Although Liu et al. (2024) introduced a $\infty$-gram LM, which extends $n$ to infinity through backoff from infinitely long $n$-grams, the effective length after backoff remains limited and cannot capture the long-range dependencies modeled by our task-gram LM. These $n$-gram LMs can be viewed as empirical approximations of the pretraining data distribution since they follow the observed frequency of $n$-grams (or pairs) in the data. Using these $n$-gram LMs, we measure the degree of memorization by correlating LLM-predicted probabilities with the probabilities generated by the $n$-gram LMs.

Our experiments focus on the Pythia model family (Biderman et al., 2023), pretrained on the Pile dataset (Gao et al., 2020), and evaluate performance across three tasks: translation (WMT (Callison-Burch et al., 2009)), factual question answering (TriviaQA (Joshi et al., 2017)), world knowledge questions (MMLU (Hendrycks et al., 2020)), and math reasoning (GSM8K (Cobbe et al., 2021)). The high-level overview of our analysis pipeline is depicted in Figure 1. We first construct a task-gram table using supervised task data, then search for the co-occurrence of $n$-gram pairs in the pretraining corpus using WIMBD (Elazar et al., 2024). Finally, we construct a task-gram LM from the co-occurrence counts and compare it with LLM predictions. Our results demonstrate that the task-gram LM effectively captures task-relevant data distributions, providing better explanations for LLM behaviors compared to the $\infty$-gram LM (Liu et al., 2024).

Among the four tasks, our analysis reveals that TriviaQA exhibits the strongest memorization effect, with task performance highly correlated to the $n$-gram distributions in the pretraining data. In contrast, MMLU shows weaker memorization, and WMT exhibits only an insignificant memorization effect. Additionally, our results show that as the model size increases, the source of performance gains varies across tasks: for TriviaQA, improved memorization plays a key role, while for MMLU, WMT, and GSM8K, increased generalization is more crucial. These findings align with the nature of the tasks: TriviaQA relies on factual recall, MMLU and GSM8K require more complex reasoning, and WMT reflects transferable translation skills.

To complement our distributional memorization analysis, we also conduct a gradient-based estimation of the training influence of pretraining documents on test examples. We find that, consistent with our distributional memorization results, pretraining documents have the largest impact on TriviaQA, followed by MMLU, and the least impact on WMT. Documents containing $n$-gram pairs from the task-gram table exert a greater influence than those containing only individual $n$-grams.

To our knowledge, this work presents one of the first comprehensive analyses of LLM capabilities by tracing their origins to pretraining corpora at scale. Our task-gram language model offers a scalable, generalizable approach for understanding LLM behavior across a variety of tasks.[2]

## 2 METHOD

Diverse abilities have been observed from LLMs trained on large pretraining corpora. Many of them are distinct in nature, like knowledge retrieval and math reasoning. It is reasonable to hypothesize that these capabilities come from different subsets of texts from pretraining. To better identify these task-relevant texts, we propose to construct a **task-gram table** from a set of supervised task data $D_T = \{(x_i, y_i)\}_i$ corresponding to task $T$, to characterize the task-relevant documents. More specifically, we mine semantically similar $n$-gram pairs $(s^x, s^y)$ from corresponding task input $x$ and output $y$ respectively. i.e., $s^x \subseteq x$, and $s^y \subseteq y$. We use the cosine similarity between the embeddings of the $n$-grams to measure the semantic closeness of the $n$-grams. The task-gram table is then constructed by all possible such $n$-gram pairs in $D_T$. The mined $n$-gram pairs capture task-specific supervision signals as the output $n$-gram can be viewed as "answering" the input $n$-gram. We formally define the task-gram table as follows:

**Definition 1.** *Denote all possible combinations of input-output $n$-gram pairs from task data $D_T = \{(x_i, y_i)\}_i$ by $A_n(T) = \cup_i [G_n(x_i) \times G_n(y_i)]$, where $G_n(\cdot)$ denotes the set of all possible $n$-grams in a piece of text. Then the **task-gram table** $H_n(T)$ is defined as:*

$$H_n(T) = \{(s_j^x, s_j^y) \mid cos(E(s_j^x), E(s_j^y)) > \gamma_T, \ s^x \neq s^y, \ (s_j^x, s_j^y) \in A_n(T)\},$$

*where $\gamma_T \in (0, 1)$ is a threshold chosen as a hyperparameter, $E$ is a pretrained text embedding model, and $cos(\cdot, \cdot)$ denotes the cosine similarity between two vectors.*

Based on the task-gram table, we can then construct a **task-gram language model**, which can describe the distribution of the characterized task-related data in a large pretraining corpus. In the following paper, we use $C$ as the counting function. We define the number of co-ocurrence of a $n$-gram pair $(s^x, s^y)$ in the same document in the pretraining copus $\mathcal{D}$ by $C((s^x, s^y), \mathcal{D})$. We also define the number of occurrences of a $n$-gram $s^y$ in the pretraining corpus $\mathcal{D}$ by $C(s^y, \mathcal{D})$. Then a task-gram language model can be defined as follows:

**Definition 2.** *A **task-gram language model** is defined over its task-gram table $H_n(T)$. For $\forall (s^x, s^y) \in H_n(T)$ and a corpus of interest $\mathcal{D}$, we define the following probability distribution*

$$P_{n,\mathcal{D}}(s^y|s^x) = C((s^x, s^y), \mathcal{D})/C(s^x, \mathcal{D}). \tag{1}$$

In practice, when we observe that an $n$-gram $s^x$ exists in an unseen text input, the chance of the corresponding $n$-gram $s^y$ appearing in the output can be estimated by $P_{n,\mathcal{D}}(s^y|s^x)$.

Suppose we have an LLM pretrained on the corpus $\mathcal{D}$. Considering the zero-shot setting where we prompt the LLM with an instruction text $u$ and an input text $x$.[3] Suppose $s^x \subseteq x$ and $s^y \subseteq y$, we

---

[2]Code will be made available upon acceptance of the paper.

[3]In practice, we use a minimal instruction template to indicate the input and output.

want to define an LLM version of the above $n$-gram conditional distribution using the LLM predicted probability of $s^y$ in the context of the concatenated testing example $u \oplus x \oplus y$:

$$P_{\text{LLM}}(s^y|s^x) = \prod_{t \in s^y} P_{\text{LLM}}(t|u \oplus x \oplus y_{[1:m-1]}). \tag{2}$$

Here $m$ denotes the location index of the $n$-gram $s^y$ found in $y$, and $t$ is each token in the tokenized $n$-gram $s^y$. We can then formally define the **distributional memorization** by Spearman correlation $\rho$ between the task-gram language model probabilities and LLM predicted probabilities of testing data:

**Definition 3.** *For a testing set $D'_T = \{(x_i, y_i)\}_i$, we denote all $n$-gram pairs found in it by $\Phi = \{(s^x, s^y)|\forall(s^x, s^y) \in [G_n(x) \times G_n(y)] \cap H_n(T), \forall(x, y) \in D'_T\}$. Then we define the extent of an **LLM distributional memorize** the pretraining corpus $\mathcal{D}$ when performing task $T$ as follows:*

$$Mem_n(LLM, \mathcal{D}|T) = \rho(\log P_{n,\mathcal{D}}(Y|X), \log P_{LLM}(Y|X)), \tag{3}$$

*where $\rho$ denotes Spearman correlation, $\log P_{n,\mathcal{D}}(Y|X) = \{\log P_{n,\mathcal{D}}(s^y|s^x)|\forall(s^x, s^y) \in \Phi\}$, and $\log P_{LLM}(Y|X) = \{\log P_{LLM}(s^y|s^x)|\forall(s^x, s^y) \in \Phi\}$.*

Similar to normal Spearman correlations, the significance of distributional generalization can be measured by $p$-value. The **distributional generalization** is then defined as the opposite of the distributional memorization: increased memorization implies decreased generalization, as the LLM predictions and the $n$-gram LM are more distributionally correlated, and vice versa.

Distributional memorization describes to what extent the LLM-predicted probability of the ground truth testing data can be viewed as a monotonic function of task-gram LM probability. This can be viewed as a measure of the predictability of task-gram LM to the LLM probability. In the following sections, we show comprehensive empirical evidence of LLMs performing knowledge-intensive tasks depending on distributional memorization while performing reasoning-intensive tasks depending on distributional generalization, as shown in Figure 2.

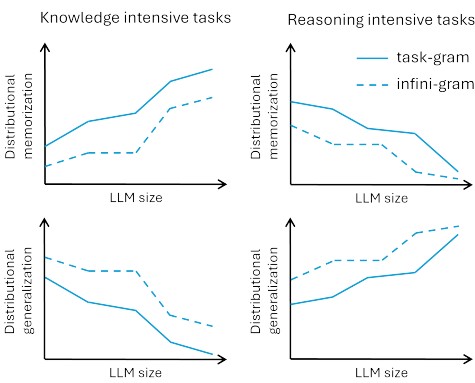

Figure 2: Expected distributional memorization and generalization trend for different types of tasks.

## 3 EXPERIMENTAL SETUP

In this section, we introduce the datasets, models, and tools we use for analyzing the memorization and generalization behaviors of LLMs.

**Models and Pretraining Corpus** We use a family of fully open-sourced, Transformer-decoder-based LMs: Pythia (Biderman et al., 2023), with a wide range of model sizes ranging from 13M to 12B parameters. All Pythia models are trained on Pile (Gao et al., 2020), a diverse pretraining corpus consisting of approximately 207B tokens. We also include some results with 1B and 7B OLMo models (Groeneveld et al., 2024), pretrained on Dolma (Soldaini et al., 2024b) with 3T tokens.

**Downstream Tasks** We use four types of tasks: machine translation, factual question answering, world knowledge questions, and reasoning.

For translation, we use the **WMT**-09 dataset (Callison-Burch et al., 2009) with a 2.5K testing set. WMT is a classic annual machine translation shared task with different languages. We chose WMT09 as our testing data instead of newer versions of WMT because WMT09 contains more European languages, which is more prominent in the Pile.

For factual question answering, we use the **TriviaQA** dataset (Joshi et al., 2017) with a 10K testing set, which is a knowledge-intensive question-answering dataset with the questions originating from trivia enthusiasts. Since the answers are usually single words or short phrases, we regard the whole answer text as the output $n$-gram $s^y$ no matter the value of $n$.

For world knowledge questions, we use the **MMLU** benchmark (Hendrycks et al., 2020), covering 57 tasks including elementary mathematics, US history, computer science, law, and more. The aim of

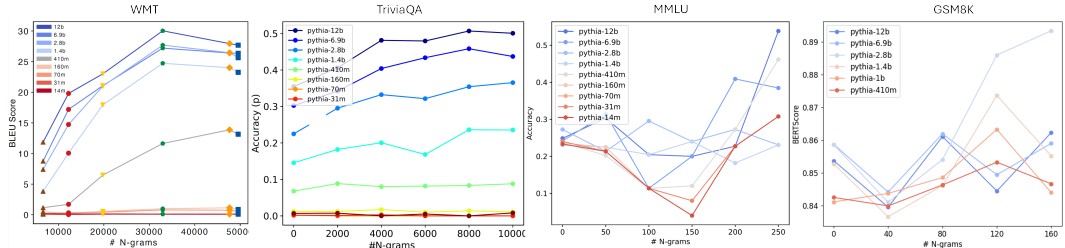

Figure 3: Task performance v.s. $n$-gram pair count in the Pile with different Pythia model sizes, for four different tasks, from left to right: WMT, TriviaQA, MMLU, and GSM8K.
For WMT, the $x$-axis shows the counts of $n$-gram pairs found in the testing set of six different languages: Hungarian, Czech, German, Italian, Spanish, and French, from left to right.

MMLU is to test models' world knowledge and problem-solving ability, so different from TriviaQA, there is a large portion of questions that require logical/math reasoning skills.

For complex reasoning, we use the **GSM8K** dataset (Cobbe et al., 2021), which contains linguistically diverse grade school math world problems with step-by-step solutions. To solve GSM8K, the LLM needs to first generate the chain-of-thoughts (CoT) reasoning step, and then draw the final conclusion, which requires advance math and logical reasoning capabilities.

**Searching over Pretraining Data at Scale**   Given the scale of the LLM pretraining corpus, searching $n$-grams over the whole corpus $\mathcal{D}$ is non-trivial. We use *What's In My Big Data?* **(WIMBD)** (Elazar et al., 2024) and the $\infty$-**gram** (Liu et al., 2024) platform, which are designed to search and retrieve documents in huge corpora. Both of them have indexed the Pile, allowing us to search it using API calls. We use WIMBD for accurately counting the co-occurrence of $n$-gram pairs as the co-occurrence frequency is usually low, and the approximate counting function in $\infty$-gram sometimes fails to capture these low-frequency $n$-gram pairs. We use $\infty$-gram for counting the occurrence of single $n$-grams as they appear more frequently in the corpus and the counting approximation has a relatively small effect on the search results. We also use the $\infty$-gram API to produce the $\infty$-gram probability of single $n$-grams, as detailed in Section 5.

## 4   $n$-GRAM DATA FREQUENCY V.S. TASK PERFORMANCE

In this section, we investigate the overall relation between the $n$-gram pair counts and the LLM task performance, which can viewed as a rough estimation of the importance of the task-related data. We confirm that the Pile is not contaminated by any of the datasets we used, by ensuring there are no large $n$-grams ($n = 8$ and $n = 14$) overlaps between the Pile and the testing data. This decontamination method is adopted from the GPT3 technical report (Brown, 2020).

We estimate the probability of a test example $(x, y)$ appearing in the pretraining corpus as the probability of any of the $n$-gram pairs from the task-gram table found in $(x, y)$ appearing:

$$P_{\mathcal{D},n}(x,y) \propto \sum_{(s^x,s^y)\in(x,y)} C((s^x,s^y),\mathcal{D})\,\mathbb{1}_{(s^x,s^y)\in H_n(T)} \qquad (4)$$

In Figure 3, we plot the task performances v.s. the count of $n$-gram pairs per example as defined in Equation (4), for WMT ($n = 2$), TriviaQA ($n = 5$), and MMLU ($n = 3$). For WMT, since the test set size for each language is the same, thus we show the sum of count per language here. For **TriviaQA**, **MMLU**, and **GSM8K**, the $x$-axis represents bins that group test examples based on the number of $n$-gram pairs. For instance, in the case of TriviaQA, data points at $x = 0$ correspond to test examples where the number of $n$-gram pairs falls between 0 and 2000. In general, TriviaQA has more $n$-gram pair counts than WMT, MMLU, and GSM8K.

The **WMT** performance is evaluated by the BLEU score between greedily generated translation and the reference translations. The **TriviaQA** performance is evaluated by the accuracy of the exact match of the generated answer. The **MMLU** performance is evaluated by the accuracy of the option

with the highest LM predicted probability. Since there are four choices for each question, the random performance of MMLU is 25%. For **GSM8K**, since the performance of Pythia models is low ($< 5\%$ accuracy), the accuracy plot is too sparse to show any trend. We compute the BERTScore (Zhang* et al., 2020) (precision) between the model-generated chain-of-thoughts (CoT) and the ground truth CoT instead.

When the model size is small ($< 410$m), **WMT** and **TriviaQA** have near-zero performance regardless of the $n$-gram pair count, while interestingly, **MMLU** reaches the lowest performance ($<10\%$) significantly lower than random guessing when the $n$-gram pair count is around 150. A closer inspection of the test examples in this interval reveals that they contain more reasoning or math problems, which appear to be harder for Pythia models. The noisier performance curve of MMLU and GSM8K is likely due to the weaker capabilities of Pythia models on this benchmark.

In general, all task performance increases when the number of task-related $n$-gram pairs increases when the model size is large enough ($> 410$m). And the trend of performance improvement is more significant when the model size is larger. For GSM8K, the Pythia 2.8B model shows the most significant increasing trend, while larger models show a less significant trend. This seems to indicate that memorization plays an essential role in LM's capabilities for all four tasks, and larger models memorize more. However, these performance curves can also be explained by the improved generalization ability of LMs when there is more relevant pretraining data. In the next section, we use pre-defined distributional memorization and generalization to investigate the possible causes behind these performance trends.

## 5 $n$-GRAM DISTRIBUTION V.S. LLM DISTRIBUTION

In this section, we compute the **distributional memorization** $\text{Mem}_n(\text{LLM}, \mathcal{D}|T)$ as defined in Definition 3, for $T = $ WMT, TriviaQA, and MMLU respectively. In addition to computing the distributional memorization with the **task-gram language model** as defined in Definition 2, we consider computing another version of **distributional memorization** with an $n$-gram language model defined by single $n$-grams. Specifically, we consider the $\infty$**-gram language model** (Liu et al., 2024) that uses an $n$ as large as possible for predicting the probability of each token, which is shown to be better aligned with human written text compared with classical $n$-gram LMs.

An $\infty$-gram LM can be viewed as an $n$-gram LM initialized with $n = \infty$, and then backoff when the $n$-gram count equals zero. This way, the probability of each token is dependent on its longest prefix that exists in the pretraining corpus. Considering concatenating the input and output text $u \oplus x \oplus y$ as the context, such distribution can be written as

$$P_{\infty,\mathcal{D}}(s^y|u \oplus x \oplus y) = \prod_{t_i \in s^y} P_{\infty,\mathcal{D}}(t_i|t_{[1:i-1]}) = \prod_{t_i \in s^y} C(t_{[i-(n_i-1):i]})/C(t_{[i-(n_i-1):i-1]}).$$

Here $i$ is the location index of the token $t_i$ in the concatenated text $u \oplus x \oplus y$, and $n_i$ is the size of the longest prefix of $t_i$ that can be found in the pretraining corpus, i.e., $n_i = \max\{n' \in [1, i] | C(t_{[i-(n'-1):i-1]}) > 0\}$. In practice, we ignore the tokens with zero probability and set the $\infty$-gram probability of this token to one, i.e., $P_{\infty,\mathcal{D}}(s^y) = 0$ only when all its tokens have zero probability. Then the alternative version of **distributional memorization** using $\infty$-gram LM is defined as:

$$\text{Mem}_\infty(\text{LLM}, \mathcal{D}|T) = \rho(\log P_{\infty,\mathcal{D}}(Y|X), \log P_{\text{LLM}}(Y|X)), \tag{5}$$

with all the notations similarly defined as in Definition 3. We then compute the two versions of distributional memorization for the different datasets, WMT, TriviaQA, and MMLU, and show the results in Figure 4.

**Translation ability does not come from memorization.** In Figure 4, we do not show any distributional memorization values of WMT because none of them are statistically significant. This indicates that while the translation performance is strongly positively correlated to the $n$-gram counts as shown in Figure 3, the performance gain does not come from initiating the pretraining data distribution.

To further investigate how different the LLM-generated translation text is from the pretraining data, we show the number of novel $n$-gram pairs that LLMs generated on WMT that have never been seen

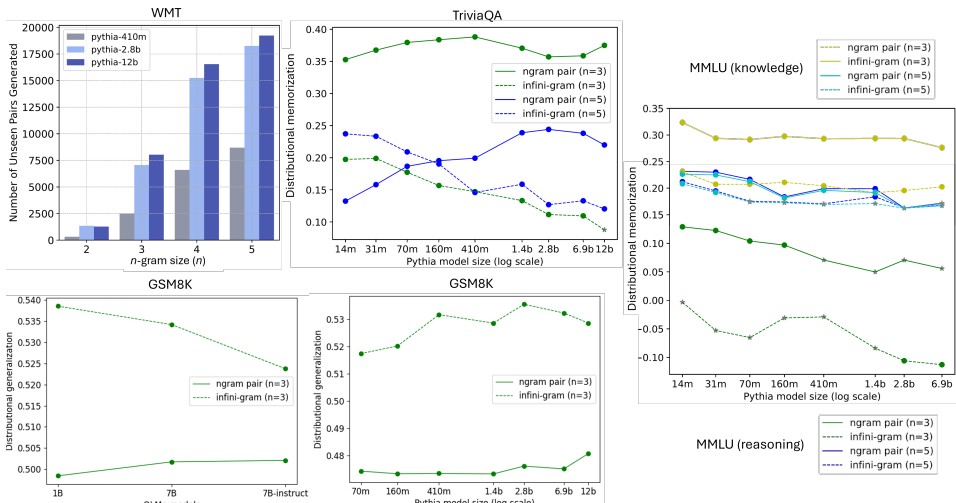

Figure 4: Visualization of distributional memorization with different-sized Pythia models on four tasks: WMT, TriviaQA, MMLU, and GSM8K. We also show results with OLMo models on GSM8K. For WMT, we show the number of new $n$-gram pairs generated by LLMs as the distributional memorization is not significant. For MMLU, we divide the tasks into two categories: knowledge-intensive and reasoning-intensive. For GSM8K, we show the Kendall tau ranking distance instead of Spearman correlation to quantify the distributional generalization effect as the distributional memorization is not significant. Solid lines show distributional memorization computed with our task-gram LM and dashed lines are computed with the $\infty$-gram LM. Statistical significant ($p < 0.05$) values are marked with solid round markers while statistically insignificant values ($p > 0.05$) are marked with gray star markers.

in any pretraining documents in the top left panel of Figure 4. It shows that larger LLMs generate more novel $n$-grams, which indicates a larger discrepancy in text distribution from the pretraining data and better distributional generalization. This implies the performance increase in Figure 3 comes from a better generalization ability learned from more relevant data instead of memorization. Our hypothesis is that this generalization is the transfer of translation skills between different languages.

Such a result contradicts the observations in Merrill et al. (2024), which show that larger LLMs are less novel in $n$-gram generation. This is likely because they evaluate LLM generation using the prompts from an in-distribution dataset (validation set of the Pile) to the pretraining corpus, which might encourage LLMs to memorize. On the other hand, the translation dataset we use is out-of-distribution, which encourages LLMs to use its generalization capabilities.

**Knowledge-intensive question-answering ability relies more on memorization.** In the top middle panel of Figure 4, we show that **TriviaQA** has a significant distributional memorization effect, in terms of both task-gram LM and $\infty$-gram LM. $\text{Mem}_{n=3}(\text{LLM}, \mathcal{D}|T)$ $(> 0.35)$ is significantly more profound than $\text{Mem}_{n=5}(\text{LLM}, \mathcal{D}|T)$ $(< 0.25)$ and $\text{Mem}_{\infty}(\text{LLM}, \mathcal{D}|T)$ $(< 0.25)$. This indicates that LLMs memorize small, long-range parallel data pieces more than large, local data pieces.

Also, when the LM size increases, our $\text{Mem}_{n=5}(\text{LLM}, \mathcal{D}|T)$ $(> 0.35)$ also increases. Since the larger model also has better performance as shown in Figure 3, this memorization behavior is highly correlated with LMs' factual QA ability. This might be because factual QA requires retrieving knowledge from training data, thus memorization plays a critical role in this task.

For **MMLU** shown in the two rightmost panels of Figure 4, we divide the 57 MMLU tasks into two groups: knowledge-intensive and reasoning-intensive. We consider knowledge-intensive tasks as tasks that can be answered by retrieving static knowledge, while reasoning-intensive tasks as ones that need computation or logical reasoning over the knowledge.[4] In general, knowledge-intensive tasks show a more significant memorization effect than reasoning-intensive tasks. This discrepancy is the most profound when $n = 3$, while the memorization level of knowledge-intensive tasks and reasoning-intensive tasks is similar when $n = 5$. This echoes our previous hypothesis that LLMs demonstrate a stronger memorization behavior when performing knowledge-intensive tasks.

---

[4]The detailed classification is shown in the appendix.

**Recalling rare knowledge requires generalization.** Similar to TriviaQA, the top right panel of Figure 4 shows that our $\text{Mem}_{n=3}(\text{LLM}, \mathcal{D}|T)$ ($> 0.25$) remains most pronounced for the knowledge-intensive MMLU tasks. The primary distinction between the MMLU tasks and TriviaQA is that $\text{Mem}_n(\text{LLM}, \mathcal{D}|T)$ decreases as the LLM size increases. This may be attributed to the fact that MMLU involves more specialized and less common knowledge compared to TriviaQA, making its occurrence in the pretraining corpus relatively infrequent. Consequently, for larger models to perform better on MMLU tasks, they may need to adjust the probability of recalling this knowledge, resulting in a decrease in distributional memorization.

**Reasoning-intensive abilities rely more on generalization.** The reasoning-intensive MMLU tasks in the bottom right panel of Figure 4 show a very different picture compared to the knowledge-intensive MMLU tasks and TriviaQA. In this case, our $\text{Mem}_{n=5}(\text{LLM}, \mathcal{D}|T)$ is the most significant, which indicates that the memorization of large text segments is more significant. This might be because some concepts can be meaningfully expressed in large text segments while the small text segments are meaningless in a reasoning-intensive context. The decreasing trend of memorization when the model size increases also indicates that memorization is not the driving force of performance improvement.

For GSM8K shown in the two bottom panels of Figure 4, we did not observe a significant memorization effect with either Pythia models or OLMo models. To quantify the distributional generalization, we substitute the Spearman correlation with the normalized Kendall tau ranking distance, which represents the fraction of data pairs that disagree on their rankings. For both Pythia models and OLMo models, the distributional generalization increases when the model size increases, while the LLMs' probabilities agree more with task-gram probabilities than the $\inf$-gram probabilities. The GSM8K results confirm that generalization is the driving force of performance improvement.

**Task-gram LM can better explain LLM predicted probabilities than $\infty$-gram LM.** With both TriviaQA and MMLU, we observe that $\text{Mem}_\infty(\text{LLM}, \mathcal{D}|T)$ is always less or equal to our $\text{Mem}_n(\text{LLM}, \mathcal{D}|T)$. And larger LLMs always show less $\infty$-gram memorization effect. This shows that our task-gram LM is a better way to model the data distribution so that it is more correlated to what is memorized by LLMs.

In general, the LLM shows decreased memorization and increased generalization when the model size increases. This implies LLMs leverage better generalization to solve hard tasks, whether they are hard in terms of the rarity of the knowledge or in terms of the requirement of reasoning. Our task-gram LM also better models the memorized data distribution than the $\infty$-gram LM.

## 6 INFLUENCE OF $n$-GRAM DATA THROUGH PRETRAINING

Since our results so far only show correlations between pretraining data frequency and the LLM predictions, we wish to explore the causal relationship of training data on LLM predictions. In this section, we estimate the influence of pretraining data on related testing predictions by accumulating the gradient dot products through model checkpoints, as introduced in Pruthi et al. (2020).

More specifically, we approximate the influence of a pretraining document at training time by the dot product between its pretraining loss gradient and the testing loss gradient. For an $n$-gram pair $(s^x, s^y) \in H_n(T)$ found in some example $(x, y)$, the test loss is defined as:

$$\ell(\theta, (x, y), s^y) = -\log P_{\text{LLM}}(s^y | u, x, y_{[1:m-1]}).$$

Here $m$ denotes the location index of the $n$-gram $s^y$ found in $y$, and $\theta$ denotes all trainable parameters of the LLM. The training loss of a pretraining document $d \in \mathcal{D}$ containing the $n$-gram pair $(s^x, s^y)$ is defined as:

$$\ell(\theta, d, s^y) = -\log P_{\text{LLM}}(s^y | d_{[1:h-1]}).$$

Here $h$ denotes the location index of the $n$-gram $s^y$ found in $d$. Suppose we have $k$ evenly spaced pretraining checkpoints of the LLM $\theta_1, \theta_2, ..., \theta_k$, then the **influence** of $d$ on the testing example $(x, y)$ through the pretraining is defined as:

$$In(d, (x, y)) = \sum_{i=1}^{k} \sum_{(s^x, s^y) \in \Phi(x,y)} \nabla_{\theta_i} \ell(\theta_i, d, s^y) \cdot \nabla_{\theta_i} \ell(\theta_i, (x, y), s^y).$$

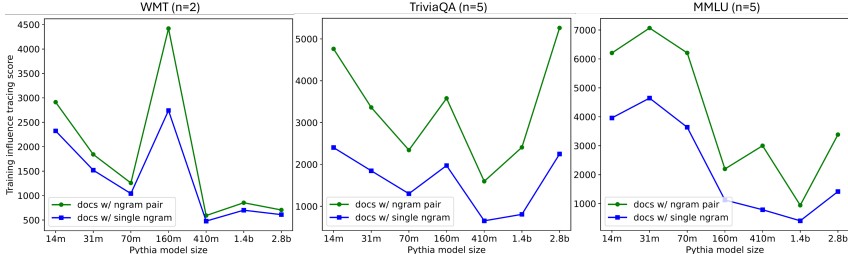

Figure 5: Training influence of pretraining documents v.s. Pythia model size with WMT, TriviaQA, and MMLU. Green lines correspond to documents containing $n$-gram pairs, while blue lines correspond to documents containing only the output $n$-gram in $n$-gram pairs.

|  | TriviaQA | | GSM8K | |
|---|---|---|---|---|
|  | Memorization | Generalization | Memorization | Generalization |
| Pythia (6.9B) | **17%** | 9% | 2.6% | **2.8%** |
| Pythia-Instruct (6.9B) | **23.5%** | 23.2% | 6.3% | **7.3%** |
| Pythia (12B) | **28.7%** | 23.2% | 2.7% | **2.8%** |
| OLMo (7B) | **36.4%** | 29.8% | 2.5% | **3.1%** |
| OLMo-instruct (7B) | **29%** | 10% | 6.3% | **7.9%** |

Table 1: Zero-shot accuracy on TriviaQA and GSM8K test set with memorization encouraged task prompt (maximize counts) and generalization encouraged task prompt (minimize counts).

Here, we use $\Phi(x, y)$ to denote all $n$-gram pairs found in $(x, y)$ and exist in the task-gram table $H_n(T)$. Such an influence function can be viewed as an estimation of the total reduction in loss on the test example $(x, y)$ that is induced by the training process whenever document $d$ is utilized in the pretraining. To estimate how much influence the documents containing $n$-gram pairs have on each testing task, we compute the average influence over the testing set $D'_T$ and $R$ randomly retrieved documents from pretraining corpus $\mathcal{D}$ for each testing example. The average influence obtained can then be written as:

$$In(\mathcal{D}, T) = \frac{1}{|D'_T|} \frac{1}{R} \sum_{(x,y) \in D'_T} \sum_{i=1}^{R} In(d_i, (x, y))$$

Note that the computation of the full gradient is relatively expensive, so we choose a relatively small $R = 50$. This analysis does not aim to cover the full pretraining corpus but to give complementary causal evidence to the previous findings. We consider two retrieval schemes: 1. retrieving documents where the test sample's $n$-gram pair appears together. 2. retrieving documents that contain the output $n$-gram from the test sample's $n$-gram pair. In Figure 5, we plot the average influence of these two schemes by green and blue curves, respectively.

Across all three datasets, we observe that pretraining documents containing $n$-gram pairs consistently contribute more to the testing examples than documents containing only the output $n$-gram in $n$-gram pairs, over different model sizes. This suggests that our task-gram table identifies important task-relevant data better than single $n$-grams. The difference between $n$-gram pair and output $n$-gram is the smallest on WMT, as well as the value of the influence function, which echoes the insignificant memorization effect of LLMs on this task. In general, when LLM size increases, WMT and MMLU decrease in data influence, while TriviaQA slightly increases, and has the highest influence value at the largest model size. This indicates that memorization is likely caused by more influence of the relevant data at training time.

## 7 PRACTICAL IMPLICATIONS: PROMPT OPTIMIZATION

An important observation of our study is that knowledge-intensive tasks benefit from LLMs' distributional memorization, while reasoning-intensive tasks benefit from LLMs' distributional generalization. Then it is possible to design or rewrite the prompt according to this principle to improve an LLM's task performance, based on the hypothesis that the LLM generation distribution is strongly affected by the prompt distribution. More specifically, to encourage memorization, we can rewrite the task instruction

to be more similar to pretraining data in terms of $n$-gram counts. To encourage generalization, we can rewrite the prompt to be less similar to training data.

We implement a simple prompt optimizer based on GPT4o and the WIMBD $n$-gram count feedback. More specifically, we instruct GPT4o (Achiam et al., 2023) to rewrite a given task prompt at each iteration, and give the average $n$-gram count in the pretraining corpus of the rewritten prompt to GPT4o in the next iteration as the reward. We instruct GPT4o to maximize this reward if we want to encourage memorization, and instruct it to minimize this reward if we want to encourage generalization. Here, we show a maximization and a minimization result for TriviaQA and GSM8K respectively. We report zero-shot testing accuracy with the Pythia models and OLMo models in Table 1. The meta prompt we used to perform such optimization and the optimized task prompts are included in Appendix E.

Note that the lengths of the optimized prompts are not significantly different, while TriviaQA significantly benefits from the prompts that are more similar to the pretraining data, and GSM8K benefits from the prompts that are less similar. More sophisticated prompt optimization algorithms with more detailed distributional memorization feedback can be designed based on a similar idea. We leave the investigation of other possibilities for future work.

## 8 RELATED WORK

**Understanding LLMs' capabilities from training data.** Most work on understanding LLMs analyzes their capabilities via synthetic experiments or small-scale studies (Arora and Goyal, 2023; Prystawski et al., 2023; Wang et al., 2024; Xie et al., 2022; Wang et al., 2023; Chan et al., 2022; Razeghi et al., 2023; Chen et al., 2024), despite the importance of scaling. Kirchenbauer et al. (2024) estimate dependencies between model capabilities and subsets of training data using kernel-based statistical evidence but rely on a small fraction of pretraining data due to computational constraints. To address this, Elazar et al. (2024) introduce WIMBD, a system for efficient $n$-gram retrieval over massive datasets, while Merrill et al. (2024) develop an unbounded $n$-gram search method and find larger LLMs generate fewer novel $n$-grams. Shaib et al. (2024) demonstrate how syntactic templates in training data influence LLM outputs, and Liu et al. (2024) propose $\infty$-gram language models to estimate text corpora distributions, focusing on local dependencies rather than complex capabilities.

In this work, we analyze the origins of LLMs' zero-shot capabilities by leveraging WIMBD and $\infty$-gram frameworks for full-scale pretraining data exploration.

**Memorization vs. generalization.** LLM memorization, defined as exact recall of training data, has been extensively studied, including memorization of rare or private data (Zhang et al., 2023), test set contamination (Jiang et al., 2024), and higher prevalence of verbatim recall in larger models (Carlini et al., 2022). Hartmann et al. (2023) provides a comprehensive summarization of different types of LLM memorization. The relationship between memorization and generalization is explored in Feldman (2020), showing that memorization can improve generalization. Extensions to this work quantify memorization via performance differences when specific training examples are included or excluded (Feldman and Zhang, 2020; Zhang et al., 2023). However, this approach is infeasible for large-scale LLMs due to retraining requirements. We propose a scalable definition of distributional memorization using $n$-gram counts to enable efficient large-scale analysis. [5]

## 9 CONCLUSION

In this paper, we introduce the **task-gram language model**, a scalable approach for modeling the task-relevant distribution of pretraining data. We trace the capabilities of LLMs back to this data by defining **distributional memorization**, measured through the Spearman correlation between task-gram LM probabilities and LLM probabilities. Through extensive experiments using Pythia models across four distinct tasks, we find that LLMs tend to memorize more when engaged in simpler, knowledge-intensive tasks, while they generalize more in harder, reasoning-intensive tasks. Our analysis provides a comprehensive examination of the origins of LLM capabilities and offers a scalable framework for investigating the fine-grained task-relevant characteristics of pretraining corpora.

---

[5]A More detailed discussion of related work can be found in Appendix C.

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

APPENDIX

## A LIMITATIONS

While this paper provides valuable insights into how large-scale pretraining corpus contributes to the emergent abilities of LLMs through $n$-gram search, there are a few limitations that we want to list out. First, the model we use, Pythia (Biderman et al., 2023), and the pretraining corpus, Pile (Gao et al., 2020), are slightly outdated and have been outperformed by many new open-source LLMs. However, most open-source LLMs lack corresponding pretraining data and have limited model sizes and pre-training checkpoints, hindering scaling effect studies. The current WIMBD system also has limitations in searching larger corpora like Dolma (Soldaini et al., 2024a) (3T tokens), calling for improved searching and retrieval methods. The quality of task-relevant $n$-gram pairs is highly sensitive to the filtering method, and while the current embedding similarity-based approach is effective, better filtering methods could significantly enhance the analysis, which is left for future research.

## B ETHICS STATEMENT

The insights and methodologies developed in this paper have several significant implications for the broader field of artificial intelligence, particularly in the development and deployment of large language models (LLMs). Understanding the balance between memorization and generalization within LLMs is crucial for both advancing the theoretical foundation of machine learning and addressing practical concerns related to their use.

**Enhanced Model Interpretability**: By extending the definition of memorization and examining how LLMs utilize their pretraining data, our research contributes to a deeper understanding of the internal mechanics of these models. This improved interpretability can help researchers and practitioners diagnose and mitigate issues related to data bias, model robustness, and unexpected behaviors in AI systems.

**Privacy and Security Considerations**: Our findings have direct implications for privacy and security in AI. Demonstrating how LLMs memorize and potentially recall training data underscores the need for rigorous data handling and anonymization techniques. It raises awareness about the risks of inadvertent leakage of sensitive information, thereby informing policy and best practices for data usage in training large models.

**Economic and Societal Impact**: As LLMs become more integral to various industries, understanding their capabilities and limitations can have significant economic and societal implications. Our research can help businesses and policymakers make informed decisions about deploying these models, ensuring they are used ethically and effectively. This, in turn, can lead to more reliable and trustworthy AI systems, fostering greater public trust and acceptance.

## C RELATED WORK DETAILS

**Understanding LLMs' capabilities from training data** Prystawski et al. (2023) and Wang et al. (2024) discuss how the reasoning ability of language models is a consequence of their pretraining data. Prystawski et al. (2023) discusses how chain-of-thought reasoning is effective in autoregressive language models because of local structure within pretraining data, and Wang et al. (2024) derives novel conclusions from known facts by aggregating reasoning paths seen in pretraining data. On the other hand, Xie et al. (2022) and Wang et al. (2023) discuss how in-context learning is a by-product of learning the pretraining data distribution. They both suggest that language models learn to implicitly infer a latent variable from the given prompt, as the pretraining data is generated from some unknown latent variable. Additionally, Chan et al. (2022) propose that the distributional properties of training data drive emergent in-context learning behaviors in large language models, whereas Razeghi et al. (2023) shows the influence the pretraining data on the mathematical abilities of LLMs. Chen et al. (2024) also highlights the significance of parallel structures in pretraining data for the emergence of in-context learning.

However, the small-scale nature of such analysis is antithetical to the commonly believed main driving factor behind the performance of LLMs: scaling. Recently, Kirchenbauer et al. (2024) proposes to provide statistical evidence of the dependence of a target model capabilities on subsets of its training data, by estimating the data distribution with an embedding-induced kernel. However, their estimation is based on a very small portion of the pretraining data (around 0.3%) as computing the embeddings of a huge dataset is very non-trivial. To get a better estimation of the whole distribution of the pretraining data, Elazar et al. (2024) construct a retrieval system, **WIMBD**, that can efficiently search $n$-gram phrases over hundreds and thousands of GBs of pretraining data. Merrill et al. (2024) propose an efficient data structure that enables unbounded-length $n$-gram searches in massive pretraining datasets, and find that larger LLMs generate less novel large $n$-grams compared to human written texts, and (Shaib et al., 2024) shows how syntactic templates from the training data causes models to re-use such templates after training. Liu et al. (2024) proposes to build a prefix-based efficient retrieval system for pretraining corpora, and then construct large-scale $\infty$-**gram** language models to estimate the distribution of text corpora. However, such distribution only models local contextual dependency, which might not be useful for understanding complex LLM capabilities.

New methods and analysis to investigate these capabilities at scale and to understand the role of scaling are needed to obtain useful insights into real-world LLMs. In this work, we aim to provide an in-depth analysis of the origin of the general zero-shot capabilities of LLMs, by performing full searches across the whole pretraining corpus with the WIMBD and $\infty$-gram framework.

**Memorization v.s. generalization** The phenomenon of machine learning models being able to perfectly memorize the training data has been studied in many previous works. Most of them define LLM memorization as exactly recalling the training examples by designed prompting, including the memorization of rare long-tail data, like private information (Zhang et al., 2023), and the contamination of testing sets (Jiang et al., 2024). Carlini et al. (2022) found that the exact copy and pasting behaviors are more prevalent in larger LMs. Hartmann et al. (2023) provides a comprehensive summarization of LLM memorization.

Several papers have studied the interplay between memorization and generalization of training data. Feldman (2020) prove that memorizing the training data is in fact required for optimal generalization on testing data. Works along this line (Feldman and Zhang, 2020; Zhang et al., 2023) extend the original definition of memorization by quantifying the extent of memorizing a training example with the performance difference when including and excluding this specific example in training data. However, this definition is impractical for large-scale analysis of LLMs as it requires retraining an LLM from scratch to analyze one data point. In this paper, we propose a new definition of distributional memorization by using $n$-gram counts, which is more suitable for large-scale analysis with LLMs.

## D EXPERIMENT DETAILS

**Dataset choice** The choice of dataset comes from the consideration of balancing different types of tasks, and we believe the combination of translation + factual QA + reasoning can well represent the spectrum of possible tasks. To enhance the selection of reasoning tasks, we also add GSM8K as a new task as described in point 1 of the general response. We found both the performance v.s. n-gram count results and the distributional generalization results on GSM8K align with our previous findings.

**Embedding model choice** The choice of embedding model comes from accommodating the need for different tasks. We use the LASER embedding model for the Translation task because it is specialized for translation. We use E5 for other tasks because it is a general-purpose sentence embedding model that performs cosine-similarity-based retrieval.

**Task-gram table construction** To build a task-gram table, we use a two-step process. First, we filter all possible input-output $n$-gram pairs in a supervised dataset by the cosine similarity between their embeddings. Second, we search through the pretraining corpus and keep $n$-gram pairs that have nonzero counts.

For WMT, we use the Europarl (Koehn, 2005) parallel corpus to construct a large task-gram table. For TriviaQA, we use its training set, while for MMLU, we use the testing set as there is no training set available. We employ two different models to filter the $n$-grams pairs. For translation tasks, we

use the LASER embeddings (Schwenk and Douze, 2017), which provide language-agnostic sentence representations, in order to assess cross-lingual similarity. We mine $n$-gram pairs from the Europarl corpus, which consists of around 20 million parallel sentences extracted from the proceedings of the European Parliament (Koehn, 2005). For all other tasks, we use E5 (Wang et al., 2022), which are multilingual sentence representations trained using contrastive learning on a diverse range of tasks. For TriviaQA, we mine the $n$-gram pairs from TriviaQA's training set. For MMLU, since the training set is very small ($100 - 500$ examples for each task), we mine the $n$-gram pairs from the test sets directly. For GSM8K, due to the short time limitation during rebuttal, we also mine the $n$-gram pairs from the 1K test sets directly.

In WMT, the cosine similarity thresholds we use are: 0.85, 0.8, 0.75, and 0.7 for 2 to 5-gram pairs, respectively; For TriviaQA and MMLU, the values are 0.75 and 0.65 for 3 and 5-gram pairs, respectively. We use lower thresholds for larger $n$-grams because larger $n$-grams inherently impose stricter alignment, and are therefore less likely. We show a cosine similarity sensitivity ablation in Figure 6.

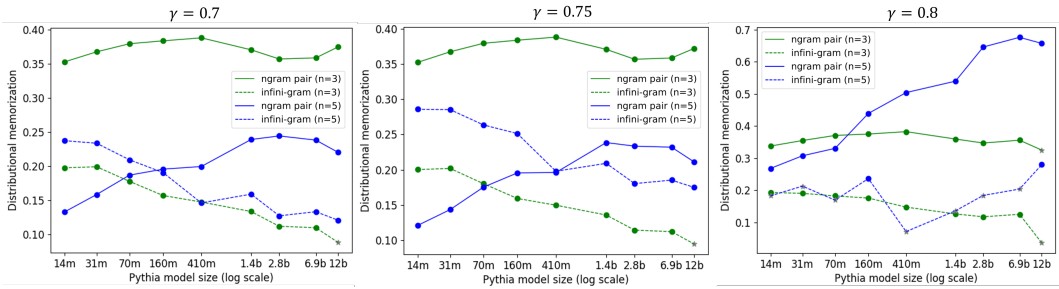

Figure 6: Visualization of distributional memorization with different-sized Pythia models on TriviaQA with different cosine similarity threshold $\gamma \in \{0.7, 0.75, 0.8\}$. The trend of distributional memorization does not change with different thresholds.

We perform our experiments on 8 GPU 40G A100 working stations. Below is the license information for the datasets we used:

- Pile: MIT license. URL: `https://github.com/EleutherAI/the-pile/tree/master`
- Tulu: ODC-BY license. URL: `https://huggingface.co/datasets/allenai/tulu-v2-sft-mixture`
- WMT-09: published with the WMT workshop. URL: `https://www.statmt.org/wmt09/translation-task.html`
- TriviaQA: Apache License 2.0. URL: `https://nlp.cs.washington.edu/triviaqa/`
- MMLU: MIT license. URL: `https://github.com/hendrycks/test`

Knowledge-intensive MMLU tasks:

```
'prehistory', 'business_ethics', 'philosophy',
'moral_disputes', 'medical_genetics', 'high_school_government_and_politics',
'human_aging', 'us_foreign_policy', 'high_school_macroeconomics',
'logical_fallacies', 'international_law', 'computer_security',
'sociology', 'professional_psychology', 'marketing', 'human_sexuality',
'anatomy', 'high_school_us_history', 'public_relations',
'high_school_microeconomics', 'clinical_knowledge', 'security_studies',
'nutrition', 'world_religions', 'high_school_psychology',
'high_school_geography', 'management', 'global_facts',
'high_school_world_history', 'high_school_european_history',
'jurisprudence', 'virology', 'astronomy', 'miscellaneous'
```

Reasoning-intensive MMLU tasks:

```
'econometrics', 'professional_law', 'abstract_algebra', 'college_medicine',
'college_chemistry', 'moral_scenarios', 'college_mathematics',
'high_school_chemistry', 'professional_accounting', 'college_computer_science',
'college_biology', 'high_school_computer_science', 'high_school_mathematics',
'college_physics', 'professional_medicine', 'elementary_mathematics',
'machine_learning', 'electrical_engineering', 'high_school_physics',
'conceptual_physics', 'high_school_statistics', 'high_school_biology'
```

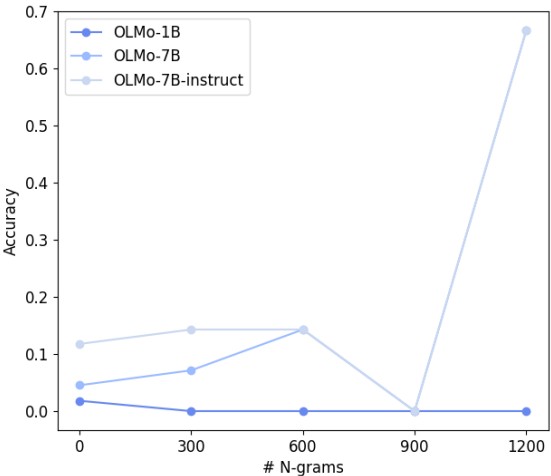

Figure 7: GSM8K accuracy v.s. $n$-gram pair count in Dolma with different OLMo model sizes.

## E  PROMPT OPTIMIZATION DETAILS

| Corpus | Dataset | Memorization | Generalization |
|--------|---------|--------------|----------------|
| Pile | **TriviaQA** | Deliver an exact single word or concise phrase in response to the factual question. (avg. 3-gram count: 557948.5) | Formulate a distinctive and concise term or phrase to clearly answer the factual question. (avg. 3-gram count: 1714.9) |
| | **GSM8K** | Carefully analyze each math word problem presented, break it down step-by-step, and clearly state the final answer. (avg. 3-gram count: 45028.9) | Dissect each math word problem into straightforward, logical steps; solve each part systematically for precise solutions. (avg. 3-gram count: 43.1) |
| Dolma | **TriviaQA** | Provide a single word or concise phrase in response to the following factual question. (Avg. 3-gram count: 5566475.5) | Provide a clear and specific word or brief phrase in response to the factual question below. (Avg. 3-gram count: 43436.2) |
| | **GSM8K** | Solve the following math word problem by methodically breaking it down into simple, clear steps to find the correct solution efficiently. (Avg. 3-gram count: 3736735.0) | Solve the upcoming math word problem by sequentially explaining each calculation and logical step, ensuring clarity and coherence in your solution. (Avg. 3-gram count: 4563.6) |

Table 2: Prompts optimized for memorization and generalization for TriviaQA and GSM8K.

Text in the square brackets are comments that are not involved in the actual prompt.

```
Meta prompt for prompt optimization

**Task Description**:

You are tasked with optimizing a given prompt to guide an open-source language model (LM)
     in completing a specific task effectively. You will receive:

- The current prompt for the task.
- Its corresponding memorization score (Average frequency of task-related n-grams found in
     the LM's pretraining corpus).
- A few example input-output pairs illustrating the intended task.
- A history of previous prompt optimization iterations.

**Optimization Goals**:

Clearly describe the intended task with a general instruction that effectively guides the
     LM to perform the task.

[If trying to encourage reasoning ...]

Minimizing the memorization score of the updated prompt. The memorization score reflects
     the distributional correlation between the prompt and the LM's pretraining corpus. A
     lower score encourages the LM to generate more novel outputs.

[If trying to encourage memorization ...]

Maximize the memorization score of the updated prompt. The memorization score reflects the
     distributional correlation between the prompt and the LM's pretraining corpus. A
     higher score encourages better alignment with the LM's learned knowledge.

**Example task input-output pairs**:

[Example input-output pairs for current task]

**Output**:

Produce an updated prompt that balances clarity of task instruction with an lower
     memorization score.
```

## F   $n$-GRAM PAIR EXAMPLES

In this section, we present some representative examples collected from the analysis for the different tasks evaluating, including Translation and Question-Answering (MMLU, TriviaQA). In order to show examples from the different experiments, we show examples with different model sizes and number of $n$-grams.

## G   MODEL GENERATIONS VS. $n$-GRAM FREQUENCY IN PRETRAINING

To further analyze the behavior of a model based on the pairs found in pretraining, we compared the alignment of the generations of the model when that $n$-gram was generated, vs. $n$-gram frequency in the pretraining corpus. We found that as the $n$-gram frequency increased, the model $n$-gram pairs the model generated were more aligned.

| Source | (English , Spanish) | Result |
|---|---|---|
| **Pythia 12b - 4gram** | | |
| The reaction of the market to the results of the vote in the American House of Representatives, which refused to support the plan for the stabilization of the financial sector there, has manifested itself here as well. | (of the vote in, de la votación en)
(the vote in the, la votación en el)
(results of the vote, resultados de la votación)
(to the results of, ante los resultados de)
(has manifested itself here, se ha manifestado aquí)
(market to the results, mercado ante los resultados)
(plan for the stabilization, plan para la estabilización)
(stabilization of the financial, estabilización del sector financiero)
(reaction of the market, La reacción del mercado) | La reacción del mercado ante los resultados de la votación en el Congreso de los Estados Unidos, que rechazó el plan para la estabilización del sector financiero allí, se ha manifestado aquí también. |
| **Pythia 410m - 4gram** | | |
| The reaction of the market to the results of the vote in the American House of Representatives, which refused to support the plan for the stabilization of the financial sector there, has manifested itself here as well. | (of the vote in, de la votación en)
(the vote in the, la votación en el)
(results of the vote, resultados de la votación)
(has manifested itself here, se ha manifestado aquí) | ' El resultado de la votación en la Cámara de Diputados, que no aceptó la propuesta de la Comisión de Desarrollo Regional para la Seguridad Social, ha manifestado aquí también.' |

Figure 8: Examples of mined pairs for the translation Task (English to Spanish) using Pythia Models with 4-Gram analysis. Models evaluated include those with 12 billion and 410 million parameters.

| Source | (English, Spanish) | Result |
|---|---|---|
| **Pythia 12b - 2gram** | | |
| The new law is also meant to constrain the "golden parachutes" for the heads of the companies participating in the program and establishes a committee which will oversee the fund. | (new law, nueva ley)
(companies participating, empresas participan)
(program establishes, programa establece)
(committee oversee, comité supervisará)
(establishes committee, establece comité)
(oversee fund, supervisará fondo)
(heads companies, jefes empresas)
(participating program, participan programa) | La nueva ley también pretende limitar los "bonos de la parrilla" para los jefes de las empresas que participan en el programa y establece un comité que supervisará el fondo. |
| **Pythia 410m - 2gram** | | |
| The new law is also meant to constrain the "golden parachutes" for the heads of the companies participating in the program and establishes a committee which will oversee the fund. | (new law, nueva ley)
(companies participating, empresas participan)
(program establishes, programa establece)
(establishes committee, establece comité)
(participating program, participan programa) | El programa de la nueva ley es también para limitar los "champios de la empresa" para los que participan en el programa y establece un comité que se encargará de gestionar el fondo. |

Figure 9: Examples of mined pairs for the translation Task (English to Spanish) using Pythia Models with 2-Gram analysis. Models evaluated include those with 12 billion and 410 million parameters.

| Question | Choices | (Key, Value) | Answer | Category |
|---|---|---|---|---|
| **Pythia 12b - 3gram** | | | | |
| These are chains of organizations that are concerned with the management of the processes and activities involved in creating and moving products from producers and manufacturers to end-user customers. The organizations involved with any one journey, are collectively termed as a: | (a) Distribution (b) Vendor (c) Communication (d) Selling | (management processes activities, Distribution) (organizations involved one, Communication) (management processes activities, Communication) (activities involved creating, Communication) (processes activities involved, Communication) | (a) Distribution | marketing |
| A 2-month-old female is brought to the office for her first routine health maintenance examination and for her immunization update. In order to determine whether or not any contraindications exist for diphtheria, tetanus, pertussis (DtaP) immunization, the parents should be questioned regarding | (a) allergy to eggs (b) Apgar scores at birth (c) gestational age at birth (d) previous seizures | (diphtheria tetanus pertussis, previous seizures) (tetanus pertussis dtap, allergy to eggs) (routine health maintenance, gestational age at birth) (pertussis dtap immunization, allergy to eggs) | (d) previous seizures | professional medicine |
| **Pythia 6.9 - 5gram** | | | | |
| 'In which of the following Asian countries would one find special economic zones (SEZs)?' | (a) Japan (b) South Korea (c) China (d) Vietnam | (find special economic zones sezs, China) (following asian countries would one, China) (asian countries would one find, China) (would one find special economic, China) | (c) China | High School Geography |

Figure 10: Examples of mined pairs for the MMLU Task using Pythia Models with bigram and 5-gram analysis. Models include 12 billion and 6.9 billion parameters.

| Question | (Key, Value) | Answer |
|---|---|---|
| **Pythia 12b - 3gram** | | |
| A tropical plant that grows several feet high, with a large fleshy root that contains an acrid milk juice, poisonous in its natural form, produces from the juice, extracted under pressure and purified, a staple foodstuff called what? | (fleshy root contains, tapioca) (tropical plant grows', tapioca) (large fleshy root, tapioca) (grows several feet, tapioca) (natural form produces, tapioca) | Tapioca |
| Feel Like Making Love and The First Time Ever I Saw Your Face were hit singles for which female artist? | (love first time, roberta flack) (time ever saw, roberta flack) (ever saw face, roberta flack) (first time ever, roberta flack) (feel like making, roberta flack) | Roberta Flack |

Figure 11: Examples of mined pairs for the TriviaQA Task using Pythia Models (12b) with trigram analysis.

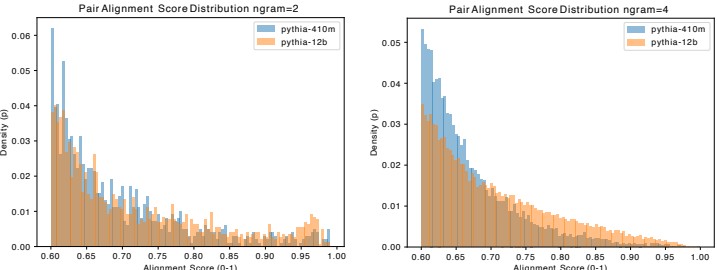

Figure 12: Model translation generations alignment for pair instances identifies in pretraining. As $n$-gram rises, the larger models are able to reproduce more aligned pairs from pretraining.

