# OpenReview forum: "Generalization v.s. Memorization: Tracing Language Models’ Capabilities Back to Pretraining Data"
_ICLR.cc/2025/Conference — ICLR 2025 Poster_

### Official Review · Reviewer_EBoK · 2024-10-27

**Soundness:** 2
**Presentation:** 2
**Contribution:** 2
**Rating:** 3
**Confidence:** 5

**Summary:**

The paper explores the debate between generalization and memorization in LLMs. It assesses how model outputs correlate with pretraining data frequencies, and proposes a task-gram language model to capture task-specific data distributions.

**Strengths:**

The paper provides a thorough investigation of the interplay between memorization and generalization in LLMs. Evaluates models on a range of tasks (translation, factual QA, reasoning), providing insights into how different tasks influence memorization and generalization.

**Weaknesses:**

1. This paper, published at the ICML 2024 Workshop on Foundation Models in the Wild, does not present significant new insights or findings. The contributions appear incremental and may not substantially advance the current understanding in the field.

2.  The conclusion that "factual question answering shows an increase in memorization with increasing LLM size" is not novel. Prior research has extensively demonstrated factual knowledge are stored in the neurons of FFNs. The increasement of factual answering task is due to having more neurons in the lareger models.

[1] Knowledge Neurons in Pretrained Transformers

[2] Transformer Feed-Forward Layers Build Predictions by Promoting Concepts in the Vocabulary Space

[3] Do Localization Methods Actually Localize Memorized Data in LLMs?

[4] Locating and editing factual associations in GPT

[5] Calibrating factual knowledge in pretrained language models

3. Lack of practical implications. It is not clear about the practical implications of its findings for improving LLM training and deployment strategies.

4. Despite covering diverse tasks, the study heavily relies on specific datasets like The Pile and utilizes Pythia models. This reliance may limit the generalizability of the findings to other datasets and model architectures.

5. The paper uses the MMLU benchmark to assess reasoning capabilities. However, MMLU is designed to evaluate a model's understanding ability across 57 subjects, including elementary mathematics, history, computer science, and law, and can not effectively reflect reasoning ability. For demonstrating reasoning skills, datasets like MGSM would be more appropriate.

6. The use of WMT-09 raises concerns due to its age and the potential for data contamination, where overlap between training and test data might exist. It is advisable to use more recent datasets, such as WMT23 or WMT24, to avoid these issues and ensure that the evaluation reflects the model's true capabilities.

7. Relying solely on Spearman correlation as a primary metric may not fully capture the nuances of memorization and generalization in language models. The paper should include additional experiments or provide mathematical justification to demonstrate why Spearman correlation is an appropriate and sufficient metric for measuring these aspects.

**Questions:**

see weakness.

---

> ### Author Response · Authors · 2024-11-22
>
> Thanks for your comments. We appreciate the reviewer found our investigation thorough with a wide range of tasks. Below, we provide a point-by-point response to your review.
>
> **Weaknesses:**
>
> **W1.** Contribution significance.
>
> We want to point out that the mentioned workshop is non-archival, and it is generally permissible to resubmit a paper accepted to a non-archival workshop. Regarding the contribution of our paper, we define a new type of memorization, beyond any existing type of memorization. Note that our conclusion of LLMs performing knowledge-intensive tasks primarily by memorization and reasoning-intensive primarily by generalization is new and has not been found by any previous work to the best of our knowledge. Our analysis presents important new evidences regarding when LLMs memorize and when they generalize. Our proposed novel analysis technique also has interesting practical implications as shown in point 4 of the general response.
>
> **W2.** Novelty compared to knowledge localization work.
>
> Thanks for bringing up the knowledge localization papers. The storage and retrieval of factual knowledge have been studied by numerous papers. However, our paper studies the interplay between generalization and memorization, which is a different topic that was not studied before, as far as we know. "Factual question answering shows an increase in memorization with increasing LLM size” is just one of our conclusions. We also draw conclusions on LLMs performing harder, reasoning-intensive tasks primarily based on generalization, which is not studied by the knowledge localization papers. Our paper also presents a novel definition of memorization, which better captures LLMs’ generalization and memorization behaviors at scale, compared to other possible alternatives like infini-gram.
>
> **W3.** Practical implication.
>
> To show the practical implication of our study, we propose a prompt optimization technique based on our observations in point 4 of our general response. Our findings suggest that LLMs’ task performance can be improved by encouraging distributional memorization or generalization depending on the task type. So we propose to use GPT4o to rewrite the task prompts and optimize memorization/generalization, which is shown to be effective on TriviaQA and GSM8K with the Pythia 12B model.
>
> **W4.** Only used Pile and Pythia.
>
> We include some new results on Dolma with GSM8K in the revision. We would love to extend our results to more pretraining corpora but most of the open-sourced LLMs trained on other corpora do not have as many model sizes as the Pythia model family, making it difficult to see the relationship between memorization and model size/task performance.
>
> **W5.** More suitable reasoning tasks.
>
> Thank you for suggesting GSM8K as a new reasoning dataset. We have included new experiments with GSM8K in the revision. For a detailed description of the new results, please see point 1 of our general response. We found both the performance v.s. n-gram count results and the distributional generalization results on GSM8K align with our previous findings.
>
> **W6.** Why use WMT09, not newer.
>
> We have performed a data contamination search with respect to all testing data, as stated in the first paragraph of Section 4. More specifically, we also perform n-gram searches with n=8 and n=14. Neither of them found any match. Checking possible testing data contamination using large n is adopted from the GPT3 technical report: they remove contaminated training texts by searching for n-gram with n=13. We chose WMT09 as our testing data instead of newer versions of WMT because WMT09 contains more European languages, which is more prominent in the Pile. Thanks for bringing this up, we have added this explanation to the revised paper.
>
> **W7.** Justification of using Spearman correlation to measure memorization.
>
> Memory and generation themselves are not well-defined: they can have a lot of different meanings. There is not a single way to quantify all possible aspects of memorization. In our paper, as stated in Definition 3, we only want to study one type of memorization: distributional memorization, which is defined by Spearman correlation. As the general memorization is not well-defined, it is not possible to provide a mathematical argument to justify such a definition.
>
> To empirically support our correlation-based analysis, in Section 6, we have shown a gradient-based experiment, which estimates the causal influence of the task-gram data throughout pretraining. In the last paragraph of this section, we discuss how these results support our correlation-based study: task-grams consistently have a higher influence than infini-gram. Translation and MMLU show decreased influence while TriviaQA shows increased influence, which has the same trend as the correlation-based distributional memorization.

---

> > ### Author Response · Authors · 2024-11-24
> > **Follow-up on Rebuttal**
> >
> > Dear reviewer,
> >
> > We just want to gently remind you that the rebuttal deadline is Nov 26, which is in two days. We are wondering if our response has adequately addressed your concerns and would be happy to clarify or discuss any remaining questions you might have.
> >
> > Best,
> > Authors

---

> > > ### Comment · Reviewer_EBoK · 2024-11-25
> > >
> > > Thank you for your clarifications.
> > >
> > > To enhance the practical implications of this method, it is necessary to provide comprehensive experimental results. These results should clearly demonstrate how the method can effectively improve the performance ofLLMs, such as GPT4o or other open-source LLMs. By substantiating these claims with empirical experiments, the potential applications and benefits of the method can be convincingly illustrated.
> > >
> > > I will keep my scores.

---

> > > > ### Author Response · Authors · 2024-11-27
> > > > **New models & clarification**
> > > >
> > > > Dear reviewer,
> > > >
> > > > As requested, we have included more models in the prompt optimization experiment in the general response and the revised paper. The new results consistently show that memorization optimized prompts benefits knowledge-intensive task (TriviaQA) while generalization optimized prompts benefits reasoning-intensive task (GSM8K).
> > > >
> > > > On the other hand, we want to clarify that the newly added prompt optimization experiment only serves as a simple illustration of **one** possible practical implication of our analysis and conclusions. The main contribution of the paper is our novel definition of distributional memorization and our pipeline to study the dynamic of LLM memorization v.s. generalization at scale. With our extensive analysis covering a wide range of tasks, we contribute an important piece of evidence regarding when LLMs memorize and when they generalize.
> > > >
> > > > We kindly request the reviewer to reconsider their evaluation of the submission.
> > > >
> > > > Best,
> > > >
> > > > Authors

---

> > > > > ### Author Response · Authors · 2024-11-30
> > > > >
> > > > > Dear reviewer,
> > > > >
> > > > > We just want to gently remind you that the rebuttal deadline is Dec 2nd, which is in two days. We are wondering if our response has adequately addressed your concerns and would be happy to clarify or discuss any remaining questions you might have.
> > > > >
> > > > > Best,
> > > > >
> > > > > Authors

---

### Official Review · Reviewer_QYo1 · 2024-11-02

**Soundness:** 3
**Presentation:** 3
**Contribution:** 2
**Rating:** 6
**Confidence:** 3

**Summary:**

The authors propose a framework for measuring the correlation between the outputs of an LLM and n-grams present in its training data.

To that purpose, they first propose a task-gram language model, which approximates the distribution of task-related data in the model's training data. They then define the distributional memorization metric (and its complementary metric, the distributional generalization) as the correlation between the probabilities of the task-gram language model and the probabilities predicted by the LLM on test data.

In the second part, the authors measure the distributional memorization/generalization of a series of LLMs of increasing size on three different tasks requiring increasing generalization capabilities. They show that the model's performance increases with the number of task-related n-grams found in the pretraining data, and that knowledge-intensive tasks rely on memorization, while reasoning-intensive tasks rely on generalization. The authors cross-check their findings by using a gradient-based method to measure the influence of training n-grams in the model outputs, which yields results that are consistent with the previous findings.

**Strengths:**

The paper focuses on the important question of the generalization-memorization tradeoff and proposes a concrete and clearly defined metric to measure it on a given LLM, which is valuable for future research.

The author's work is well-written and structured, clear, and easy to read. The experimental methodology is sound, with some possible flaws and limitations described in the appendix.

I appreciate the fact that the authors confirmed their findings using the gradient-based method and that they checked for possible cross-contamination of the training dataset.

**Weaknesses:**

My main concern regarding this work is that the distributional memorization/generalization metrics depend on a large number of hyperparameters (choice of dataset and embedding model, cosine similarity thresholds), but although the authors justify some of the choices they made regarding those, they do not measure how sensitive the metrics are to the value of those parameters. While I understand that the proposed method is computationally expensive, I believe that this needs to be addressed to assess the relevancy and usefulness of the proposed metric and strengthen the author's claims. In addition, some choices are unclear. For example, different values are picked for the 5-gram threshold choices for WMT compared to the other datasets, but this is not justified.

A smaller point of criticism is that the authors define distributional generalization as the divergence between the distribution of the LLM’s output and that of the pretraining data. While necessary, this criterion is insufficient: for example, smaller LLMs tend to fall into degenerate/repetitive distributions which do not conform to natural language syntax, and would therefore obtain a high generalization score. Performing fluency measurements or qualitative discussion of the results would help with this aspect.

An editing mistake seems to appear in the paper in Def. 3 : "we define the extent of an LLM distributional memorize the pretraining corpus".

**Questions:**

- Appendix: The example n-grams in Figure 5 appear to contain stop words, while those in Figures 6-8 do not. Could the authors clarify whether stop word removal was performed when mining n-gram pairs? Since this seems to be the most computationally expensive part of this work, limiting the number of obtained pairs may help with scaling. More generally, my intuition is that systematic enumeration of all n-gram pairs is probably superfluous. Have the authors considered performing NER-based filtering post-mining?

- In Figure 2, the smaller models are depicted to obtain a ~10% accuracy on MMLU, which is below random chance, when the number of N-grams in the training set is 150. Do the authors have any qualitative insight on why this may be happening?

- The Pile was checked for potential overlaps with the training datasets using exact n-gram matches with n > 14. As this value is quite large, especially if stopwords were removed, could the authors explain why this particular value was chosen?

---

> ### Author Response · Authors · 2024-11-22
>
> Thanks for your positive comments! We appreciate that the reviewer found the question we studied important and our paper “structured, clear, and easy to read”. The reviewer also found our experimental methodology sound, and liked that we confirmed our findings with a gradient-based method and checked possible test set contamination. Below, we provide a point-by-point response to your review.
>
> **Weaknesses:**
>
> **W1.** Hyperparameter choice and sensitivity.
>
> We chose the cosine similarity threshold based on the human inspection of the task-gram quality (i.e. how well they represent the current task). We use smaller thresholds for larger n-grams to tolerate more noises contained in them. We have added a cosine similarity threshold sensitivity experiment in Figure 5 in the Appendix as described in point 3 of the general response. **The trend of distributional memorization does not change across different cosine similarity thresholds** (0.7, 0.75, and 0.8), and our method is in general robust to the choice of cosine similarity threshold. The 0.05 difference between WMT and other datasets is only a result of human preference, which does not affect the final conclusion.
>
> The choice of dataset comes from the consideration of balancing different types of tasks, and we believe the combination of translation + factual QA + reasoning can well represent the spectrum of possible tasks. To enhance the selection of reasoning tasks, we also add GSM8K as a new task as described in point 1 of the general response. We found both the performance v.s. n-gram count results and the distributional generalization results on GSM8K align with our previous findings.
>
> The choice of embedding model comes from accommodating the need for different tasks. We use the LASER embedding model for the Translation task because it is specialized for translation. We use E5 for other tasks because it is a general-purpose sentence embedding model that performs cosine-similarity-based retrieval.
>
> Thanks for bringing this up. We have added the above discussion to Appendix D.
>
> **W2.** The possible influence of generation fluency.
>
> The LLM predicted probability is not evaluated on LLM-generated outputs, but on the ground truth outputs, thus does not have the fluency issue. The LLM-generated outputs are used for evaluating the task performance and counting the number of novel/unseen n-gram pairs.
>
> **Questions:**
>
> **Q1.** Stop word filtering and NER-based post-filtering.
>
> For the main paper results, we only perform stop word filtering with Translation tasks, as the translation of stop words in European languages is very similar and can result in identical pairs. For other tasks, we do not perform stop word filtering. In Appendix F, we showcase the matched task-gram pairs in Pythia model generations, while we mine these pairs from the ground truth data. For displaying per-instance results we used both techniques interchangeably, just to showcase their occurence in the pretraining data.
>
> For additional filtering of the mined n-gram pairs, we have not performed any such (e.g. NER based) post-filtering. Due to the short time constraint of the rebuttal period, we are not able to extensively modify and rerun our pipeline at this time, but we would like to investigate this factor after the rebuttal.
>
> **Q2.** MMLU performance drops at 150 frequency.
>
> Yes. As stated in the second last paragraph of Section 4: “A closer inspection of the test examples in this interval reveals that they contain more reasoning or math problems, which appear to be harder for Pythia models.” More specifically, this interval contains more n-grams with numbers and math operations. The questions containing these n-grams likely contain math operations.
>
> **Q3.** Choice of large n-gram for decontamination.
>
> For contamination check, we do not remove stop words. In addition to n=14, we also perform a check with n=8. Neither of them found any match. Checking possible testing data contamination using large n is adopted from the GPT3 technical report: they remove contaminated training texts by searching for n-gram with n=13.

---

> > ### Comment · Reviewer_QYo1 · 2024-11-25
> >
> > I thank the authors for their clarifications and answers, as well as for the new appendices. I have no further questions or comments at this point.

---

> > > ### Author Response · Authors · 2024-11-27
> > >
> > > Dear reviewer,
> > >
> > > Thanks for your response. We would appreciate it if you could consider raising your score in light of the new experiments and clarifications we have provided.
> > >
> > > Best,
> > >
> > > Authors

---

### Official Review · Reviewer_w9Gw · 2024-11-05

**Soundness:** 4
**Presentation:** 4
**Contribution:** 3
**Rating:** 8
**Confidence:** 4

**Summary:**

The paper adds a thorough analysis of memorization vs generalization capability of LLMs taking into account the pretraining data. Using the Pythia model family and the Pile dataset, the paper proposes a task-based n-gram language model, which is built on downstream tasks, by extracting semantically related n-grams. This n-gram model is then used to compute correlation among seen/novel n-grams with the LLM output perplexity, to come up with a notion of distributional generalization. More correlated these metrics are (task n-gram vs LLM), the bigger the amount of memorization and vice-versa. The key contribution of the paper is this methodological approach, using which they show clear memorization and generalization aspect based on different downstream tasks.

**Strengths:**

- The paper is exceptionally well written - I found it very enjoyable to read it end to end.
- The methodological contribution is strong and makes sense to compute correlation between task specific n-grams and LLM perplexity. This method can likely be extended to multiple different tasks easily, with the caveat of the availability of the pretraining data distribution.
- While memorization vs generalization debate has a long list of papers in the literature, this paper makes an important contribution by computing the statistics _at scale_ by considering full pretraining data distributions.
- The contributions builds upon relevant literature ($\infty$ gram and WIMBD).
- Good choice of downstream tasks tested, ranging from translation, to factual to math.

**Weaknesses:**

- Perhaps some more downstream datasets could have been considered apart from MMLU, as it is known the dataset contains several annotation issues [1].
- The argument of MMLU requires more reasoning than generalization due to math is also not completely answerable. It would be interesting to dive into that question using GSM8k, given recent results point more to memorization issues [2].
- Not much of a weakness per se, but a limitation (which the authors duly acknowledge) that the results are conditional on the Pile dataset.


[1] https://arxiv.org/pdf/2406.04127v2
[2] https://arxiv.org/abs/2410.05229

**Questions:**

- In the definition of task-gram language model, what happens when the corresponding n-gram of the denominator is rare or absent? I assume you filtered them out from the computation, but wondering what is the % of n-grams you had to drop per dataset.
- How does your setup account for the synonyms of the n-grams? This is probably going to be more of an issue for smaller datasets such as MMLU, and can likely be a confound to the results?
- In your opinion, does the varying amount of samples in test datasets pose as a confound? Since MMLU is the smallest dataset, perhaps one experiment could be done by taking bootstraps from WMT and TriviaQA in the same sample sizes?
- Not sure I follow the reasoning behind using *both* $\infty$-gram and WIMBD. Wouldn't this add a bias? Curious how plots change by using only one or the other.
- For TriviaQA results, any insight on why does $\infty$-gram reduces with scale?
- Here is a hypothetical scenario: suppose if you add the task data progressively into the LLM training (by finetuning), then the task-n-gram probability can be controlled from 0 to max smoothly. Since this is finetuning, the model will be more incentivized to memorize the data aggressively. What kind of distributional generalization curves you would expect to see?


## Suggestions

- From a cursory glance i missed that Figure 3 scales of MMLU knowledge and reasoning are different, which is important to highlight in my opinion. I understand making a single axes removes the ability to show the trend (stuff will be a straight line), but if there is a way to visually highlight the range of scales it would be great.
- Section 2 can have a small plot showing the expected trend of memorization vs generalization, to ground the readers how to interpret the later plots.
- Typo: L499 - ~pertaining~ -> pretraining

---

> ### Author Response · Authors · 2024-11-22
>
> Thank you for your positive comments! We truly appreciate that the reviewer found our paper to be “exceptionally well written” and “very enjoyable to read it end to end”. The reviewer also found our “methodological contribution is strong” and our “method can likely be extended to multiple different tasks easily”. Comparing with existing literature, the reviewer found our paper “makes an important contribution by computing the statistics at scale”. The reviewer also found we have a “good choice of downstream tasks”. Below, we have provided a point-by-point response to your review.
>
> **Weaknesses:**
>
> **W1&2.** More downstream tasks.
>
> Thank you for suggesting GSM8K as a new reasoning dataset. We have included new experiments with GSM8K in the revision. For a detailed description of the new results, please see point 1 of our general response.
>
> **W3.** Only use Pile.
>
> We include some new results on Dolma with GSM8K in the revision. However, since OLMo only has two model sizes, we can only show results with these two sizes. We would love to extend our results to more pretraining corpora but most of the open-sourced LLMs trained on other corpora do not have as many model sizes as the Pythia model family, making it difficult to see the relationship between memorization and model size/task performance.
>
> **Questions:**
>
> **Q1.** Percentage of zero denominator.
>
> The percentage of n-gram pairs that we need to drop due to the zero denominator is relatively small, around 5-10%, because single n-grams are usually easy to find in Pile.
>
> **Q2.** Synonym n-grams.
>
> We do observe the existence of synonyms in mined n-gram pairs, especially in MMLU and GSM8K, likely because many of the input and output texts are related logically instead of semantically. However, some synonyms are potentially meaningful, for example, the same phrase in question form and not in question form. We currently do not have a filter for synonyms, but we would like to investigate this factor after the rebuttal as there is not enough time to modify and rerun the searches before the rebuttal ends.
>
> **Q3.** Testing sample size as a confounder.
>
> Since the distributional memorization is calculated on an n-gram pairs level, and usually an n-gram pair only appears once in the test set, the effective example number should be the number of non-zero count n-gram pairs, which is actually similar for MMLU and TriviaQA. We will add an experiment with the same number of n-gram pairs for all datasets.
>
> **Q4.** Why use both infini-gram and WIMBD.
>
> We use infini-gram for counting the denominator mainly because it is much faster than WIMBD, which enables faster experiment iteration. The occurrence of single n-grams is more frequent and thus can tolerate the counting approximation used in infini-gram. The cooccurrence of n-gram pairs is rare and infini-gram’s approximation ends up overlooking them, thus we have to use the full search of WIMBD.
>
> **Q5.** Different memorization trends between infini-gram and task-gram.
>
> The infini-gram only produces probability for single n-grams, while the task-gram produces conditional probability for task-related n-gram pairs. The infini-gram reducing with scale for TriviaQA reflects that larger LMs do not plainly memorize the local probability distribution, but a task-relevant long-range probability distribution.
>
> **Q6.** Progressively adding task-related data.
>
> The shape of the distributional memorization curve would depend on the effect of the task-related data: whether its effect of helping the LM to generalize to other unseen cases is stronger or its effect of grounding the LM to memorize its distribution is stronger. The accurate answer can only be confirmed with experiments, but from the existing observations, we expect that knowledge-intensive tasks should display increasing memorization, while reasoning-intensive tasks should display increasing generalization. From our current results, the task-related data for reasoning-intensive tasks seem to help the task performance by encouraging LMs to produce more novel generations, while for knowledge-intensive tasks, the task-relate data encourages memorization.
>
> **Suggestions:**
>
> **S1:** Thanks for pointing this out. We have combined the MMLU figures together under the same scale in the revised paper.
>
> **S2:** Thanks for the suggestion. We have added such a plot in the revision.

---

### Official Review · Reviewer_YUoe · 2024-11-06

**Soundness:** 3
**Presentation:** 3
**Contribution:** 2
**Rating:** 3
**Confidence:** 3

**Summary:**

This paper proposes a "task-gram language model," built on n-gram co-occurrences, to trace task-specific data in pretraining corpora. Through experiments on translation, question answering, and reasoning tasks using Pythia models, the paper finds that memorization is predominant in simpler, fact-based tasks like factual question answering, while generalization is more critical for complex tasks requiring reasoning.

**Strengths:**

- The proposed task-gram model offers an alternative approach to evaluating LLM behavior by connecting output distributions directly with pretraining data.
- Experimental results reveal several interesting findings like how model size influences memorization versus generalization, showing that larger models tend to generalize more in complex tasks, an insight useful for understanding scaling effects.

**Weaknesses:**

- The task-gram model can be insufficient in capturing the complexity of today's generative tasks with short input descriptions. These tasks often demand high levels of abstraction and creativity from the LLMs to infer extensive information from minimal data, a process that cannot be well represented through n-gram frequency metrics. I recommend evaluating on more complex tasks like code generation (e.g., HumanEval) and math reasoning (e.g., GSM8K).
- Using Spearman correlation between task-gram and LLM probabilities may conflate memorization with necessary task handling where n-gram overlap is essential. For example, high correlation in translation could reflect accurate language pairing rather than mere memorization. This might misinterpret critical task-related patterns as memorization.
- Overall, the findings indicate that different tasks exhibit varying levels of dependence on memorization, which is not novel or surprising, as already discussed in previous works like [1].
- Several methodological and experimental designs need clarification. See questions for details.

[1] Hartmann, Valentin, Anshuman Suri, Vincent Bindschaedler, David Evans, Shruti Tople, and Robert West. "Sok: Memorization in general-purpose large language models." arXiv preprint arXiv:2310.18362 (2023).

**Questions:**

1. In Figure 2, how are n-gram occurrences counted when a sentence contains multiple task-related n-grams? Specifically, are overlapping n-grams counted individually or merged, and how does this approach impact the interpretation of memorization levels in longer and more complex sentences?
2. For generative tasks with very short input descriptions, is this approach still applicable, and how does it handle sparse n-gram distributions?
3. How does the methodology account for task-related patterns that are critical for accuracy rather than memorization, such as translation?
4. The study shows that in knowledge-intensive MMLU tasks, memorization decreases as model size increases. Could this decrease be due to specific limitations in capturing rarer knowledge from larger models, or might it indicate a shift toward more generalized, inferential capabilities?
5. Appendix E states that alternative methods (e.g., NER models) for identifying n-gram pairs yielded inconsistent results. Could you elaborate more about your implementation and the results?
6. The experiment uses different cosine similarity thresholds across tasks (WMT v.s. TriviaQA and MMLU). How did you select these hyperparameters, and how sensitive are the results to these thresholds?
7. Some tasks might share similar patterns or data structures (e.g., QA and factual retrieval), could memorization from one task inadvertently support generalization in another? Does the study address potential cross-task memorization effects, especially for overlapping or semantically similar tasks in the training data? The task-specific n-gram approach may suffice to take these cross-task effects into account.
8. Beyond the analysis and findings, can you propose a potential method for actively adjusting the balance between memorization and generalization within models to improve performance across tasks?

---

> ### Author Response · Authors · 2024-11-22
>
> Thanks for your comments. We appreciate that the reviewer found our finding “an insight useful for understanding scaling effects”. Below, we provide a point-by-point response to your review.
>
> **Weaknesses:**
>
> **W1**. Task-gram sufficiency in capturing complex tasks.
>
> While task-gram is not as complex as LLMs, we found it strikes a good balance between expressiveness of the task and faithfulness to the pretraining data. Our extensive empirical analysis also shows that task-gram can explain a good portion of LLMs’ task performance by distributional memorization or generalization. As suggested by the reviewer, we have added a new experiment on GSM8K, which is detailed in the general response and the revised script.
>
> **W2**. Memorization v.s. task features.
>
> The task-grams mined by our method are indeed essential to the task performance, as shown in Figure 2 of the paper, that all task performance is positively correlated with more task-gram counts. However, distributional memorization as defined in Definition 3 with Spearman correlation, does not necessarily positively correlate with the task performance. In Figure 3, Translation and GSM8K do not show a significant distributional memorization effect. MMLU shows a negative correlation between task performance (implied by the model size) and distributional memorization. In summary, our results show that distributional memorization does not correlate with task performance.
>
> **W3**. Related work discussing similar topics.
>
> Thanks for pointing out the related work, which is a nice position/survey paper that proposes a taxonomy that summarizes all existing LLM memorization works together. We have cited it in our related work section in lines 516-517. Note that our paper defines a new type of memorization, beyond any type of memorization discussed in the reference paper. Note that our conclusion of LLMs performing knowledge-intensive tasks primarily by memorization and reasoning-intensive primarily by generalization is new and has not been concluded by any previous work, to the best of our knowledge. Whether LLMs primarily memorize or generalize when performing different tasks still does not have a good consensus in the community. Our analysis presents important new evidence regarding when LLMs memorize and when they generalize. Our proposed novel analysis technique also has interesting practical implications as shown in point 4 of the general response.

---

> ### Author Response · Authors · 2024-11-22
>
> **Questions:**
>
> **Q1**. n-gram merging.
>
> The overlapping n-grams are counted individually. Note that the distributional memorization is computed on a n-gram level instead of on an example/sentence level, so the probabilities of individual n-grams do not need to be merged. Thus, the text length does not impact the memorization score computed for the current task.
>
> **Q2**. Short task input description.
>
> We assume that the input description referred to by the reviewers means the task instruction. In our experiments, we only use task descriptions within one or two words, like “Question: “. Also, the task instruction is not involved in mining the task-gram pairs, only the raw task input-outputs are used. We do not see how short input descriptions would impact our method. On the other hand, very short task inputs or outputs that are smaller than the used n-gram would have an impact. In this case, we use the whole input or output as the n-gram if their length is smaller than n. For the sparse n-gram distribution issue, we only use the task-gram pairs with non-zero counts to compute the distribution memorization score. As shown in point 2 of the general response, there are in general enough task-grams for each task.
>
> **Q3**. Memorization v.s. Task-related patterns.
>
> Please see our response to W2. While the task-gram pairs counts are indeed important for task performance as shown in Figure 2, our proposed distributional memorization does not correlate with task performance.
>
> **Q4**. Decreased ability in capturing rare knowledge v.s. Better generalization.
>
> We conclude that the decreased distributional memorization in MMLU implies better generalization because the performance is also increased for larger models. It is not reasonable if both memorization and generalization decreased but the performance increased.
>
> **Q5**. Alternative method in Appendix.
>
> The scope and setting of the project were very different from our current paper in our early exploration. This alternative method is not used for identifying parallel n-gram structures, but for identifying any general parallel pieces. In this alternative method, we prompted GPT3.5 or fine-tuned Llama with a pretraining document and its entity tags produced by another NER model, and instructed the model to classify whether this document contains any “parallel structure”. Our initial hypothesis was that the general “parallel structure” presented in the pretraining data, like QA or translation-like text pieces, can be viewed as a form of implicit supervision of the LLMs, which is the reason why the LLMs can perform diverse tasks in zero-shot. However, since the “parallel structure” itself is not well-defined, our attempt to identify it failed.
>
> **Q6**. Choice cosine similarity threshold.
>
> We chose the cosine similarity threshold based on the human inspection of the task-gram quality (i.e. how well they represent the current task). We use smaller thresholds for larger n-grams to tolerate more noises contained in them. We have added a cosine similarity threshold sensitivity experiment in Figure 5 in the Appendix as described in point 3 of the general response. The trend of distributional memorization does not change across different cosine similarity thresholds, and our method is in general robust to the choice of cosine similarity threshold.
>
> **Q7**. Cross-task memorization.
>
> We primarily focus on the dynamic between memorization and generalization within the same task in this paper. While the cross-task memorization effect is an interesting direction that our proposed method can be used to study, we leave it to future research due to space and time limitations.
>
> **Q8**. Practical implication.
>
> Yes. We propose a prompt optimization technique based on our study in point 4 of our general response. we propose to use GPT4o to rewrite the task prompts and optimize memorization/generalization, which is shown to be effective on TriviaQA and GSM8K with the Pythia 12B model.

---

> > ### Author Response · Authors · 2024-11-24
> > **Follow-up on Rebuttal**
> >
> > Dear reviewer,
> >
> > We just want to gently remind you that the rebuttal deadline is Nov 26, which is in two days. We are wondering if our response has adequately addressed your concerns and would be happy to clarify or discuss any remaining questions you might have.
> >
> > Best,
> > Authors

---

> > > ### Author Response · Authors · 2024-11-27
> > >
> > > Dear reviewer,
> > >
> > > We would be happy to discuss and provide further clarification if you have any additional questions. Otherwise, we kindly ask you to reconsider your score in light of the new experiments and clarifications we have provided.
> > >
> > > Best,
> > >
> > > Authors

---

> > > > ### Author Response · Authors · 2024-11-30
> > > >
> > > > Dear reviewer,
> > > >
> > > > We just want to gently remind you that the rebuttal deadline is Dec 2nd, which is in two days. We are wondering if our response has adequately addressed your concerns and would be happy to clarify or discuss any remaining questions you might have.
> > > >
> > > > Best,
> > > >
> > > > Authors

---

### Author Response · Authors · 2024-11-22
**General Author Response**

We thank all reviewers for their insightful comments. In the general response, we would like to present some new experiment results and statistics. Detailed findings are included in the revised manuscript, with newly added content highlighted in blue. We will continue refining both the experiments and the manuscript following the rebuttal.

**1. New task and new pretraining corpus:**

We added a new math reasoning task, GSM8K, which contains more complex reasoning. For GSM8K, we show both results with the Pile (Pythia models), and Dolma (OLMo models). Since OLMo models only have two sizes 1B and 7B, we also experiment with a 7B OLMo model instruction tuned on Tulu. Due to the short time constraint, we only show n=3 for the n-gram size for this experiment at this time. We have ensured that Tulu does not contain any task-gram pairs with n=3.

The GSM8K performance v.s. n-gram pair count in the Pile plot is shown in Figure 2 in the paper. Since the performance of Pythia models is in general low (<5% accuracy), the accuracy plot is too sparse to show any trend. We thus compute the BERTScore (precision) between the model-generated CoT and the ground truth CoT instead. GSM8K in general follows the same trend that performance increases when the number of task-related n-gram pairs increases when the model size is large enough. Similar to MMLU, GSM8K shows a noisier trend, which is likely related to their reasoning-intensive nature: LLMs are less dependent on distributional memorization. The plot shows that the Pythia 2.8B model shows the most significant increasing trend, while larger models show a less significant trend. This also can be explained by the decreased distributional memorization. We show the GSM8K accuracy v.s. n-gram pair count in the Dolma with OLMo models in Figure 6 in the Appendix.

As expected, the distribution memorization effect, as defined in Definition 3, of GSM8K is not significant (p > 0.05), so we show the Kendall Tau distance between the LLM probability and the task-gram probability to quantify the generalization effect. A normalized Kendall Tau distance represents the fraction of data pairs that disagree on their rankings. In Figure 3, we show the Kendall Tau distance for both Pythia models and OLMo models.  For both Pythia models and OLMo models, the distributional generalization increases when the model size increases, while the LLMs’ probabilities agree more with task-gram probabilities than the infini-gram probabilities. This agrees with the trend observed with MMLU, and supports our hypothesis that LLMs’ reasoning-intensive abilities rely more on generalization, and that task-gram LM can better explain LLM-predicted probabilities than infini-gram LM.

**2. n-gram pair count sparsity statistics:**

There is a significant portion of the task-gram pairs that cannot find any match in the pretraining corpus. However, decreasing the value of n for n-grams can significantly help with this issue. In the table below, we show the number of task-grams we found in the Pile with non-zero counts. There are in general enough task-gram pairs for each task.

| |TriviaQA | MMLU | Translation |
| ---- | ---- | ---- | ---- |
| n=3 | 88514 | 73776  | 12552 |
| n=5 |12386 | 7291 | 1159 |

**3. Cosine similarity threshold sensitivity ablation:**

To build a task-gram language model, we need to first construct a task-gram table that consists of semantically relevant n-gram pairs minded from task input-output pairs. In our pipeline, we set a cosine similarity threshold of the n-gram’s embeddings to filter task-gram pairs. We chose the cosine similarity threshold based on the human inspection of the task-gram quality (i.e. how well they represent the current task).

In Figure 5 in the Appendix, we show the cosine similarity threshold sensitivity of distributional memorization with TriviaQA, by varying the similarity threshold between 0.7, 0.75, and 0.8. As we can see, the trend of increased memorization with increased model size does not change across different cosine similarity thresholds.

---

> ### Author Response · Authors · 2024-11-22
>
> **4. Practical implication:**
>
> An important observation of our study is that knowledge-intensive tasks benefit from LLMs’ distributional memorization, while reasoning-intensive tasks benefit from LLMs’ distributional generalization. Then it is possible to design or rewrite the prompt according to this principle to improve an LLM’s task performance, based on the hypothesis that the LLM generation distribution is strongly affected by the prompt distribution. More specifically, to encourage memorization, we can rewrite the task instruction to be more similar to pretraining data in terms of n-gram counts. To encourage generalization, we can rewrite the prompt to be less similar to training data.
>
> We implement a simple prompt optimizer based on GPT4o and the WIMBD n-gram count feedback. More specifically, we instruct GPT4o to rewrite a given task prompt at each iteration, and give the average n-gram count in the pretraining corpus of the rewritten prompt to GPT4o in the next iteration as the reward. We instruct GPT4o to maximize this reward if we want to encourage memorization, and instruct it to minimize this reward if we want to encourage generalization. Here, we show a maximization and a minimization result for TriviaQA and GSM8K respectively. We report zero-shot testing accuracy with the Pythia models and OLMo models below. The meta prompt we used to perform such optimization is included in Appendix E.
>
> | Model | TriviaQA Memorization | TriviaQA Generalization | GSM8K Memorization | GSM8K Generalization |
> |---------------------------|------------------------|--------------------------|---------------------|-----------------------|
> | Pythia (6.9B) | **17%** | 9% | 2.6% | **2.8%** |
> | Pythia-Instruct (6.9B) | **23.5%** | 23.2% | 6.3% | **7.3%** |
> | Pythia (12B) | **28.7%** | 23.2% | 2.7% | **2.8%** |
> | OLMo (7B) | **36.4%** | 29.8% | 2.5% | **3.1%** |
> | OLMo-Instruct (7B) | **29%** | 10% | 6.3% | **7.9%** |
>
> We also show the optimized prompts below:
>
> | **Corpus** | **Dataset** | **Memorization** | **Generalization** |
> |------------|-------------|-------------------|---------------------|
> | **Pile**   | **TriviaQA** | Deliver an exact single word or concise phrase in response to the factual question. (avg. 3-gram count: 557948.5) | Formulate a distinctive and concise term or phrase to clearly answer the factual question. (avg. 3-gram count: 1714.9) |
> |            | **GSM8K**    | Carefully analyze each math word problem presented, break it down step-by-step, and clearly state the final answer. (avg. 3-gram count: 45028.9) | Dissect each math word problem into straightforward, logical steps; solve each part systematically for precise solutions. (avg. 3-gram count: 43.1) |
> | **Dolma**  | **TriviaQA** | Provide a single word or concise phrase in response to the following factual question. (avg. 3-gram count: 5566475.5) | Provide a clear and specific word or brief phrase in response to the factual question below. (avg. 3-gram count: 43436.2) |
> |            | **GSM8K**    | Solve the following math word problem by methodically breaking it down into simple, clear steps to find the correct solution efficiently. (avg. 3-gram count: 3736735.0) | Solve the upcoming math word problem by sequentially explaining each calculation and logical step, ensuring clarity and coherence in your solution. (avg. 3-gram count: 4563.6) |
>
> Note that the lengths of the optimized prompts are not significantly different, while TriviaQA significantly benefits from the prompts that are more similar to the pretraining data, and GSM8K benefits from the prompts that are less similar. More sophisticated prompt optimization algorithms with more detailed distributional memorization feedback can be designed based on a similar idea. We leave the investigation of other possibilities for future work.

---

### Meta-Review · Area_Chair_6H3p · 2024-12-14

**Metareview:**

The paper provides a novel methodology for analyzing memorization vs generalization in LLMs across diverse set of tasks. Also agree with reviewer w9GW that this method can likely be extended to multiple different tasks and overall introduce a useful methodology for probing LLMs. Agree with some concerns raised by the reviewers that some of the findings are not surprising or new necessarily but the scale and level of rigor provided in this work make it an important contribution to the understanding of LLMs inner-working.

**Additional Comments On Reviewer Discussion:**

The authors have successfully replied to the reviewers and also added missing pieces to the paper.

---

### Decision · Program_Chairs · 2025-01-22

Accept (Poster)